# Atomic and Molecular Laser-Induced Breakdown Spectroscopy of Selected Pharmaceuticals

**Pravin Kumar Tiwari** [1,2], **Nilesh Kumar Rai** [3], **Rohit Kumar** [3], **Christian G. Parigger** [4] and **Awadhesh Kumar Rai** [2,*]

1   Institute for Plasma Research, Gandhinagar, Gujarat 382428, India
2   Laser Spectroscopy Research Laboratory, Department of Physics, University of Allahabad, Prayagraj 211002, India
3   CMP Degree College, Department of Physics, University of Allahabad, Pragyagraj 211002, India
4   Physics and Astronomy Department, University of Tennessee, University of Tennessee Space Institute, Center for Laser Applications, 411 B.H. Goethert Parkway, Tullahoma, TN 37388-9700, USA
*   Correspondence: awadheshkrai@rediffmail.com; Tel.: +91-532-2460993

**Abstract:** Laser-induced breakdown spectroscopy (LIBS) of pharmaceutical drugs that contain paracetamol was investigated in air and argon atmospheres. The characteristic neutral and ionic spectral lines of various elements and molecular signatures of CN violet and $C_2$ Swan band systems were observed. The relative hardness of all drug samples was measured as well. Principal component analysis, a multivariate method, was applied in the data analysis for demarcation purposes of the drug samples. The CN violet and $C_2$ Swan spectral radiances were investigated for evaluation of a possible correlation of the chemical and molecular structures of the pharmaceuticals. Complementary Raman and Fourier-transform-infrared spectroscopies were used to record the molecular spectra of the drug samples. The application of the above techniques for drug screening are important for the identification and mitigation of drugs that contain additives that may cause adverse side-effects.

**Keywords:** paracetamol; laser-induced breakdown spectroscopy; cyanide; carbon swan bands; principal component analysis; Raman spectroscopy; Fourier-transform-infrared spectroscopy

---

## 1. Introduction

Paracetamol (PCM) is a medication that is frequently used for the relief of mild to moderate pain and fever experienced by people of all ages [1]. It is also registered in the model list of essential medicines published by the World Health Organization (WHO) that communicates the most important medications for the sustenance of human health [2]. However, routine use of PCM may cause complications that could potentially cause liver damage, and simultaneously, a decrease of pain thresholds [2]. The pharmaceuticals would serve their intended purpose only if they are free from impurity or other interference that might be harmful for human health. In addition, various chemical and instrumental methods are regularly introduced in the pharmaceutical drug industry for the evaluation of drug contra-indications and effects on the human body [1,2].

At present, various laser-based approaches are available for the elemental analysis of solids, liquids, gases, and heterogeneous biological matrices including drug samples; but fast and cost-effective techniques are vital in manufacturing [3,4]. During the last few decades, laser-induced breakdown spectroscopy (LIBS) has been successful as a reliable, first-choice spectroscopy and analytical technique in various fields of applications [5–9]. Recent studies reveal that LIBS can be an efficient tool for the rapid identification and quantification of a drug's elemental composition [7–10].

In the optical emission spectroscopy reported in this work, namely LIBS, high peak-irradiance radiation of the order of 100 GW/cm$^2$ is applied for the initiation of breakdown at and near the surface of the target material. It is a minimally destructive technique for determining the composition of the material in any phase (solid, liquid, and gas) with little or no sample preparation. It offers in-situ, online and real-time operations for the materials, even those that are remotely located [7–10].

LIBS, is in general viable for elemental analysis, and conditionally, LIBS is also being used to predict the presence of molecules in the sample [11–16]. The molecular emission is generally more complex than that involving atomic emission. Local plasma conditions, as well as plasma cooling duration, may have an effective role in the changes in the measured radiation originating from molecules [7–9]. The present manuscript communicates the analysis of drugs such as PCM that are organic in nature. In the presence of organic molecules in the sample, the molecular signature of CN violet and C$_2$ Swan bands could be observed in the LIBS spectra [9,11–14]. The fingerprint of CN and C$_2$ molecular bands in the LIBS spectra of organic compounds have been examined to correlate the spectral molecular emission from the laser-induced plasma and the molecular structure of an organic compound present in the sample [11–16].

Other spectroscopic techniques, called Raman and FT-IR spectroscopy, are complementary to LIBS and have also been used for the study of molecules present in drug samples. Raman spectroscopy is a vibrational spectroscopic technique that provides molecular structural information about the materials without any sample preparation [17,18]. The absorption spectrum of the drug sample has also been recorded using FT-IR spectroscopy [18].

The hardness of medical pills is usually not mentioned in pharmacopeia specifications [19–22]. However, to produce a quality product it is essential to decide the upper and lower limits of the hardness of drugs at the time of manufacturing. Too soft tablets can disintegrate in transport and too hard tablets could cause several complexities in bioavailability, for example, the hardness of chewable tablets should be suitable [22]. There are different approaches to measure the hardness of a drug sample, and one of the recent approaches is using LIBS [19,20]. The hardness of a sample can be predicted using the ionic to atomic line ratio of an element present in the LIBS spectra. In the present manuscript, the relative hardness of drug samples has also been evaluated.

In the present work, six different brands of pharmaceutical pain-reliever drugs (Table 1) have been studied using LIBS in two different environments (air and argon). A statistical approach, namely, principal component analysis (PCA) [10] has also been applied to classify the drug samples. The present study infers that LIBS could be a more practical approach for the online performance analysis of drug samples.

**Table 1.** Sample details along with molecular formula and total weight % of carbon and nitrogen.

| Sample | Compound in Sample (Manufacturer) | % of C | % of N |
|--------|-----------------------------------|--------|--------|
| S1 | Ibuprofen C$_{13}$H$_{18}$O$_2$ (400 mg) and Paracetamol C$_8$H$_9$NO$_2$ (325 mg) | 50.9 | 30.1 |
| S2 | Aceclofenac C$_{16}$H$_{13}$Cl$_2$NO$_4$ (100 mg) Paracetamol C$_8$H$_9$NO$_2$ (500 mg) | 37.2 | 50.2 |
| S3 | Diclofenac Sodium C$_{14}$H$_{11}$Cl$_2$NNaO$_2$ (50 mg) Paracetamol C$_8$H$_9$NO$_2$ (325 mg) | 23.2 | 32.3 |
| S4 | Paracetamol C$_8$H$_9$NO$_2$ (500 mg) | 31.7 | 46.3 |
| S5 | Paracetamol C$_8$H$_9$NO$_2$ (500 mg) | 31.7 | 46.3 |
| S6 | Ibuprofen C$_{13}$H$_{18}$O$_2$ (400 mg) | 30.2 | 0 |

## 2. Results

### 2.1. LIBS Analysis

The LIBS spectra of the drugs, summarized in Table 1, have been recorded in ambient atmosphere. The observed atomic and ionic emissions corresponding to the organic elements are C (247.8 nm), H$\alpha$ (656.3 nm), O (777.4 nm), and N triplet lines (742.46, 744.28, 746.85 nm) are depicted in Figure 1a. The spectral lines of the inorganic elements, such as Na (589.0, 589.5 nm), Mg (279.5 (II), 280.2 (II), 285.2, 382.9(II), 383.2(II), 516.7, 517.3, 518.3 nm), Ca (393.3(II), 396.8(II), 397.2(II), 422.6 nm), and Si (250.7, 251.4, 251.6, 252.8, and 288.1 nm), have also been observed in the LIBS spectra of these drugs [23].

Figures 1–3 display the molecular spectra corresponding to the CN violet system (B$^2\Sigma^+$–X$^2\Sigma^+$) and C$_2$ Swan band system (d$^3\Pi$g–a$^3\Pi$u), observed in the LIBS spectra of these drugs. The spectra are expected to be formed by recombination of native carbon–carbon and carbon–nitrogen in the laser-induced plasma of the drug samples [13–16,24,25]. However, another possibility of the appearance of CN features in LIBS spectra may be due to the interaction between atmospheric nitrogen and the laser-induced plasma. Therefore, to suppress atmospheric interference, LIBS spectra of these drugs have also been recorded in an argon atmosphere. The experiments reveal CN and C$_2$ bands in LIBS spectra of these drug recorded in argon atmosphere. This confirms that the origin of the CN band in the LIBS spectra of the drug is likely (primarily) due to recombination of native carbon-carbon and carbon-nitrogen in the laser-induced plasma of the drug (Figure 1b,c, Figures 2b and 3a).

The LIBS spectra of the investigated drugs contain $\Delta v = 0$ and $\Delta v = 1$ sequences of CN and C$_2$ band systems. Figures 1a and 2a shows the presence of CN violet bands in the argon atmosphere at 388.2, 387.0, 386.1, 385.4, and 385.0 nm corresponding to (0,0), (1,1), (2,2), (3,3) and (4,4) vibrational transitions, respectively, in samples S1–S5, but these bands are absent in sample S6, which contains iburophen, but no paracetamol [26]. Similarly, Figures 1c and 3a reveals that the spectral peaks of a C$_2$ Swan band system are observed in samples S1–S6 at 516.4 and 512.8 nm correspond to (0,0) and (1,1) bands. In addition to the (0,0) and (1,1) vibrational bands of C$_2$, the molecular bands at 471.5, 469.7, 468.4, and 467.8 nm correspond to the (2,1), (3,2), (4,3), (5,4) bands that are observed in the LIBS spectra (Figure 1c).

If there are no C–C, or C = C bonds in the molecular structure of the sample, then C$_2$ bands will not be observed in the LIBS spectra of the sample [15]. Therefore, to predict the signature of the molecule in the sample, the strongest emission bands of C$_2$ (0,0) system at 516.4 nm and CN (0,0) system at 388.32 nm are included in calculations, rather than using a carbon line. An analysis of these bands that are recorded with sufficient spectral resolution contains information about the molecule, i.e., it establishes the signature of the molecule present in the sample as described in Table 1.

The variation observed in the spectral intensity of CN and C$_2$ bands in the LIBS spectra represents the presence of different kinds of molecules in the matrix of pharmaceutical samples. In addition to this, with the help of the reported molecular formula of the drugs, the carbon and nitrogen percentages were calculated and correlated with the CN and C$_2$ vibrational band intensities present in the LIBS spectra of the corresponding drugs (Table 1).

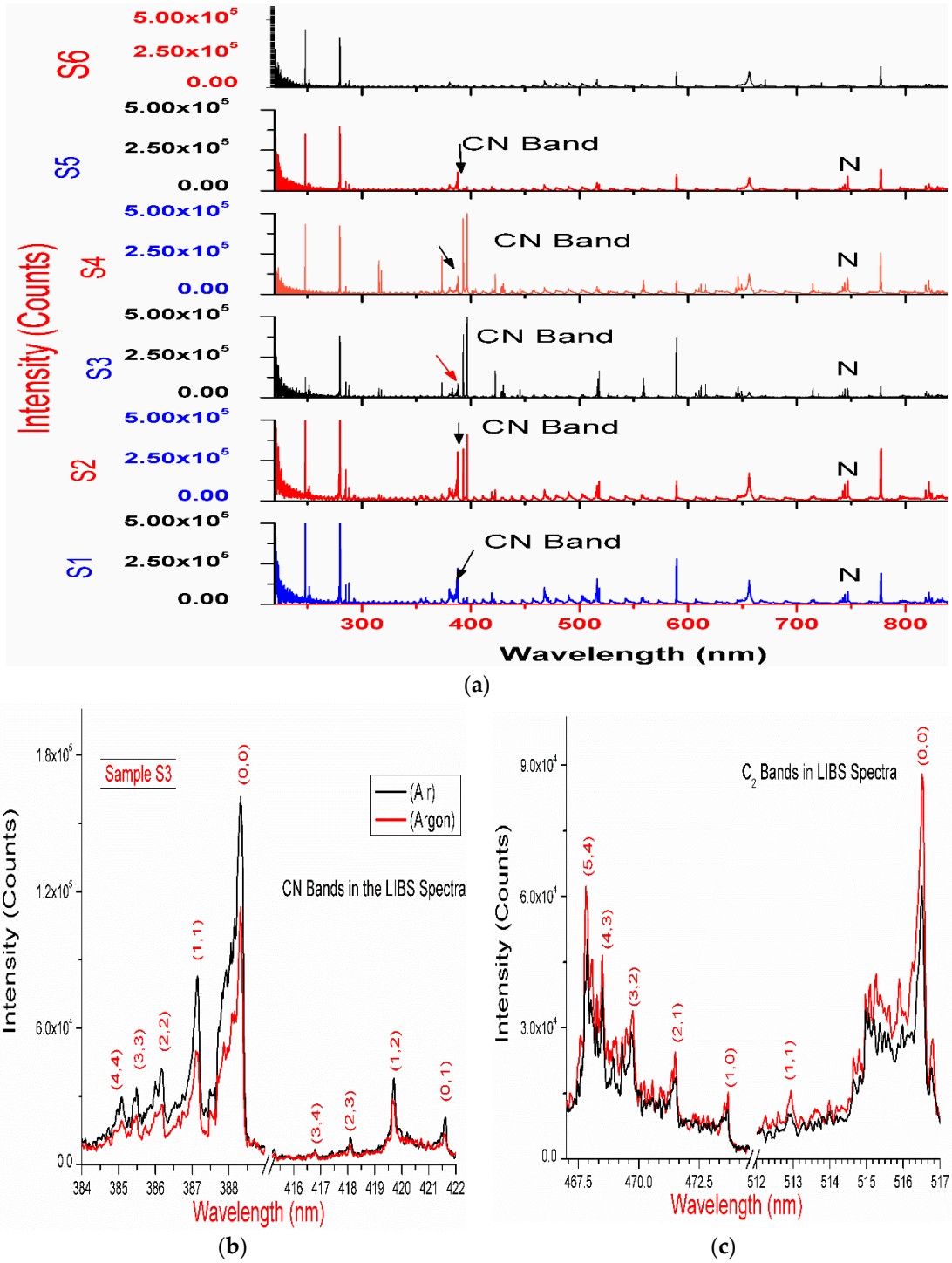

**Figure 1.** (**a**) The laser-induced breakdown spectroscopy (LIBS) spectra of drugs S1–S6 recorded in ambient atmosphere, 240–850 nm; (**b**,**c**) LIBS spectra (average of 10 spectra of each spectra having 50 accumulations) of sample S3 recorded in argon and ambient atmosphere.

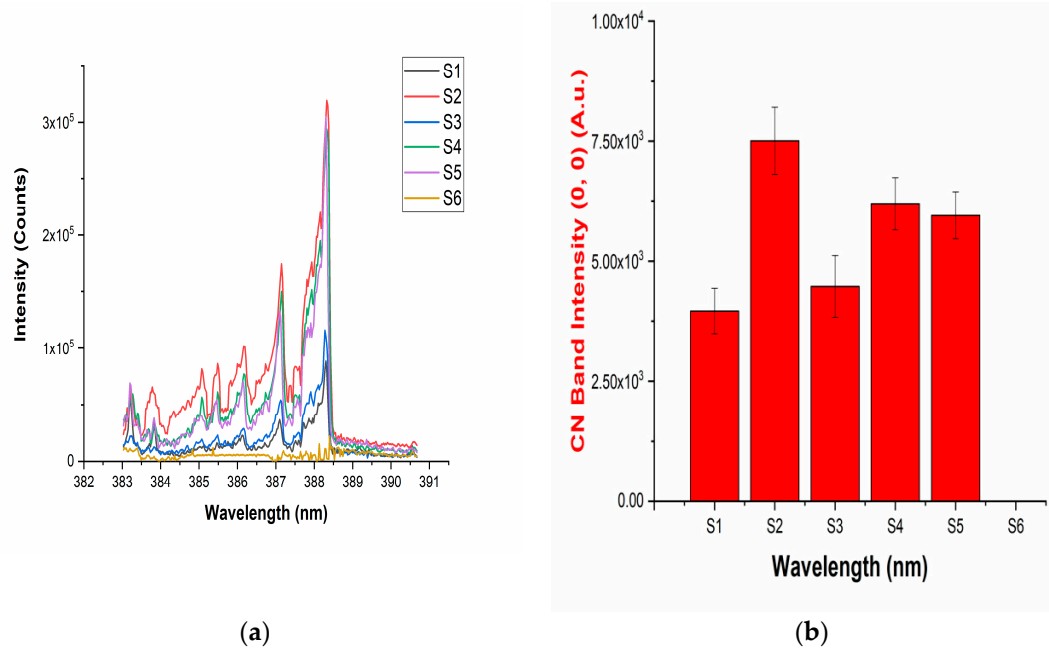

(a)
(b)

**Figure 2.** (**a**) The LIBS spectra of the drugs showing the presence of the vibrational band of a CN molecule; (**b**) intensities of the (0,0) band of the CN band present in the LIBS spectra of the drugs.

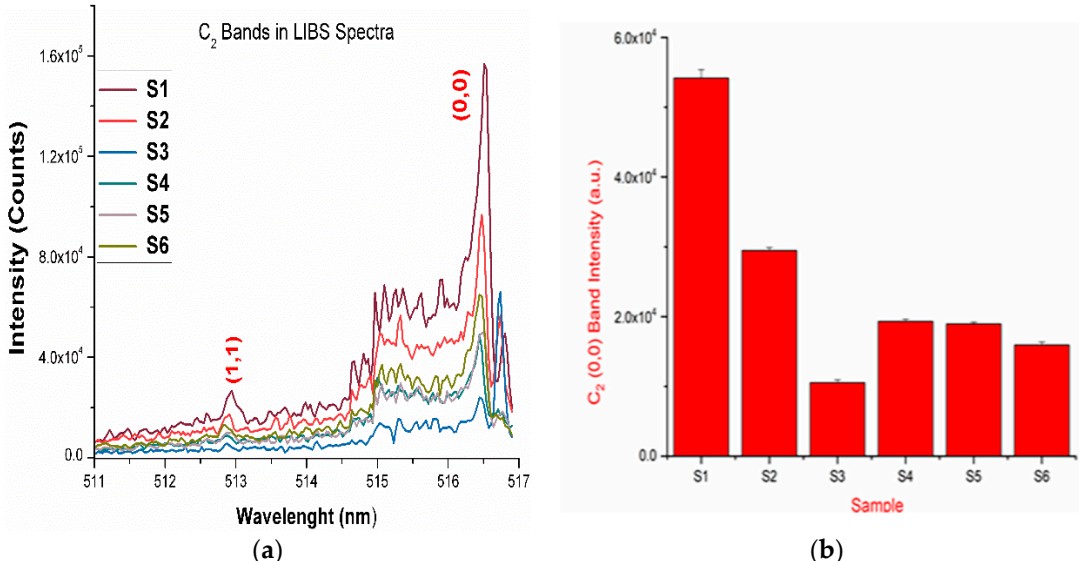

(a)
(b)

**Figure 3.** (**a**) The LIBS spectra of the drugs showing the presence of the vibrational band of a $C_2$ molecule; (**b**) intensities of the (0,0) band of the C2 band present in the LIBS spectra of the drugs.

### 2.2. Relative Hardness of the Tested Drugs

Using the LIBS, the relative hardness of the drugs (tablets) was measured. The hardness of the material can be evaluated from the intensity ratio of the 373.6 nm ionic line of Ca II to that of the neutral 422.6 nm line of Ca I. The higher the ratio, the harder the sample [19]. Alternatively, the intensity ratio of the ionic spectral lines Mg II at 279.5 nm to that of the neutral lines Mg I at 285.2 nm can also be used to determine the hardness of the tablet [20]. The emission line intensity alone cannot be used to determine the hardness of the tablet because matrix effects and a change of the experimental parameters with a sample can affect the intensities [19,20].

The interference-free emission line of calcium has been taken for the calculation of the intensity ratio. The measurement is repeated for the ratio of ionic lines to neutral lines of magnesium observed

in the LIBS spectra of the drugs belonging to different brands and the results are shown in Figure 4a,b. It is observed that the ratio changes with respect to the sample and calculated data suggest that sample S2 and S4 are harder in comparison to all other samples. Therefore, the density of the drugs in terms of the hardness of the drug has been calculated. This methodology can be significant for monitoring the uniform density of the tablet at the time of manufacturing and batch performance analysis of the drug.

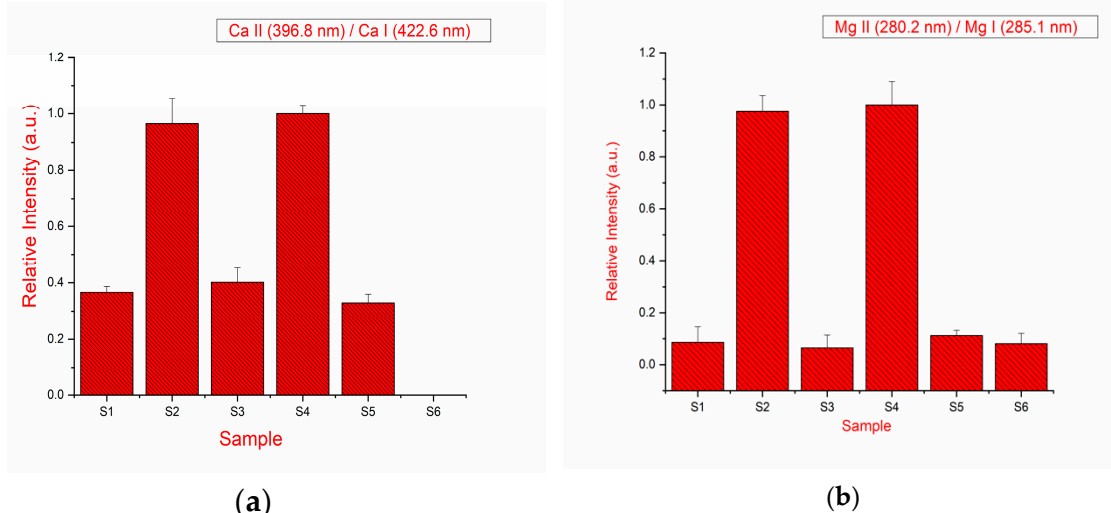

**(a)**        **(b)**

**Figure 4.** (**a**) Intensity ratio of spectral lines Ca (II) 396.8 nm and Ca (I) 422.6 nm; (**b**) intensity ratio of spectral lines Mg (II) 279.5 nm and Mg (I) 285.2 nm.

### 2.3. Principal Component Analysis (PCA)

The LIBS spectra of samples S1–S6 are shown in Figure 1a. Qualitatively, it is clear from Figure 1a that there could be slight variations in the spectral signature and thus elemental composition among all the drug samples. However, even for slight variations in the composition of the drug samples, the discrepancy among the six types of the drug was investigated with a PCA of the LIBS spectral datasets. The principal component scatter-plot was drawn using the LIBS spectral datasets of samples S1–S6 recorded in an argon atmosphere to represent the three-dimensional PCA. The graph represents one point in Figure 5 for each spectrum in terms of the principle components (PCs). PCA gives the visual representation of the dataset through projection. One can extract important information from the variables (in the form of principal components) of the dataset. The uncorrelated variables in the PCA are known as principal components [27]. The library set of the LIBS spectra was represented in the form of the matrix of certain order ($60 \times 24,806$) for the six drug samples of different brands. Using Unscrambler-X software [10], the matrices were transformed into principle components (PCs) representations. Weighted linear combinations of the variables were found to describe the major trends in the data. The representations in PCA are displayed in score plots, namely in terms of PCs of different dimensions including PC-1, PC-2, and PC-3.

Figure 5 depicts three different PCs (PC-1, PC-2, and PC-3) to elucidate the variations of the spectra as well as their scores for discrimination of the samples. In the experiments, 10 LIBS spectra were recorded for each brand of the sample and these data were included for the PCA analysis. Apparent clustering of each brand sample can be noticed in Figure 5, which shows obvious differences between the six groups of drug samples. The sample of S1 (blue data), S2 (red data), S3 (dark green data), S4 (black data), S5 (light green data) and S6 (brown data) in Figure 5 are color-coded. Samples S4 and S5 are closely clustered to each other, which indicates that both the sample may have common compositions. It is also mentioned in Table 1 that S4 and S5 have similar compositions. The discrimination observed in the PCA score plots for the data recorded in ambient air is close to that in the argon atmosphere. The analysis shows that information extracted from PCA allows significant classification of the drugs despite them having similar compositions.

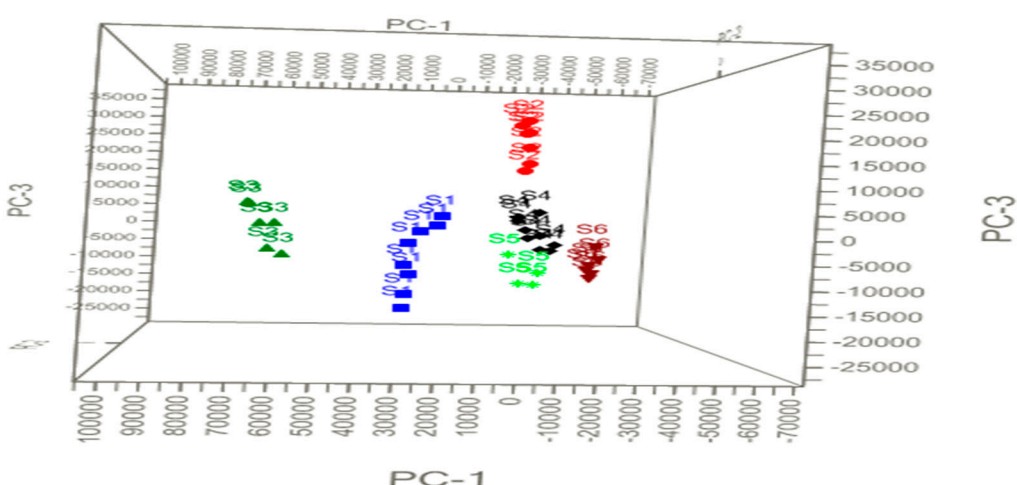

**Figure 5.** Principle component scatter (PCA) plot of the drugs in three dimensions. PC-2 axis is perpendicular to the plane of this page.

*2.4. Raman Spectroscopy*

Raman spectra of drugs have been recorded for identification and verification of molecular composition present in the drug. The PCA plot in Figure 5 reveals that sample S3 is highly discriminated from the other samples, and therefore, we have taken sample S1 and S3 for Raman analysis. The observed peaks in the Raman spectra of samples S1 and S3 are tabulated in Table 2 and depicted in Figure 6a,b.

Analysis using Raman characteristic group frequencies [18] indicates that the peaks at 3049 cm$^{-1}$ could be identified as associated with C-H stretching mode; and the peaks at 1647 and 1611 cm$^{-1}$ are associated with C = O stretch. The vibrational mode with a peak at 1566 cm$^{-1}$ is associated with N-H bending and the mode with a peak at 1115 cm$^{-1}$ is associated with aromatic ring breathing. The vibrational mode with peaks at 801 cm$^{-1}$ and 750 cm$^{-1}$ are found to be associate with C-H bending (para) and C-N stretching mode, respectively. The vibrational modes around 2800–3000 cm$^{-1}$ are associated with alkyl C-H bands [18,25]. With the present resolution, most of the vibrational modes observed in the Raman spectra of these samples are common (see Table 2) which confirms the similar functional group of the samples.

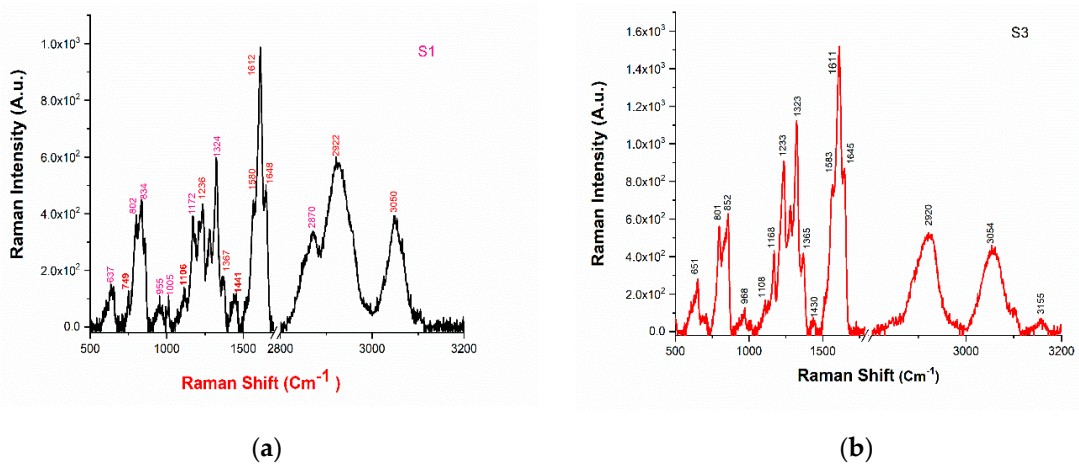

(a)　　　　　　　　　　　　　　(b)

**Figure 6.** (**a**) The Raman spectrum of sample S1; (**b**) the Raman spectrum of sample S3.

**Table 2.** Observed vibrational modes in the Raman spectra of sample S1 and S3.

| Sample S1 Raman Shift (cm$^{-1}$) | Sample S3 Raman Shift (cm$^{-1}$) | Result |
|---|---|---|
| 749 | —— | C-N stretch |
| 802 | 801 | C-H Bend, para |
| 1116 | —— | Aromatic ring breathing |
| 1213 | 1211 | C-O Stretch |
| 1236 | 1230 | C-O Stretch |
| 1367 | 1365 | CH$_3$ bend |
| 1441 | —— | O-H bend |
| 1580 | 1583 | N-H Bend |
| 1612 | 1611 | C=O stretch |
| 1648 | 1645 | C=O stretch+ N-H deformation |
| 2927 | 2920 | Saturated C-H |
| 3050 | 3055 | Aromatic C-H |

*2.5. FT-IR Spectroscopic Analysis*

The FT-IR spectra of sample S1 and S3 are shown in Figure 7a,b. The distinctive vibrational peaks at 3320–3335 cm$^{-1}$ were assigned to NH stretching. Vibrational peaks at 1650–1730 cm$^{-1}$ and 1575–1560 cm$^{-1}$ were attributed to C = O stretching, and C = C stretching, respectively. Absorption bands at 2955 cm$^{-1}$ represents C-H stretching whereas bands at 1422, and 1380 cm$^{-1}$ were assigned to C-H bending; and vibrational peaks at 780 cm$^{-1}$ represent fingerprint stretching of C-Cl. Furthermore, the Raman peaks at wavenumber 1712, 935 and 780 cm$^{-1}$ are not observed in the FT-IR spectra of sample S1. The presence of these additional vibrational peaks represents the signature of an additional functional group, i.e., diclofenac sodium in sample S3 [28].

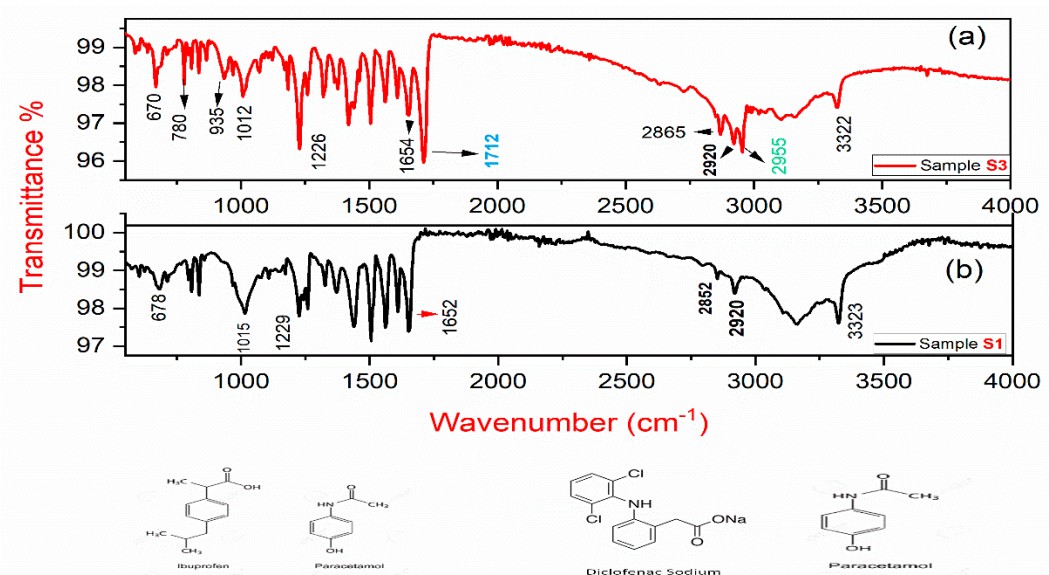

**Figure 7.** (**a**) FT-IR spectrum of sample S3; (**b**) the FT-IR spectrum of sample S1.

Thus, our present molecular study infers that the molecular study using the LIBS technique confirms the presence of different molecules in samples S1–S6, which stems from a complementary analysis using Raman and FT-IR spectroscopies. One can conclude that LIBS allows one to study elemental as well as molecular compositions.

## 3. Materials and Methods

In our LIBS set-up as described elsewhere [10], high peak-power and frequency-doubled 532 nm, nanosecond pulses from a Nd:YAG laser were used to generate plasma at the sample surface. The laser beam was focused onto the sample surface using a convex lens of 15 cm focal length. The repetition rate of the laser pulses was kept at 2 Hz with 15 mJ per pulse of laser energy. The emitted signals from the cooling plasma were recorded. The spectra show neutral and ionic atomic lines together with molecular bands [7,9]. For resolving the atomic spectral emissions from the ablated sample, the emissions from plasma were dispersed with a Mechelle spectrometer (spectral range 200 nm to 900 nm, $\lambda/\Delta\lambda \approx 6000$; Mechelle ME5000, Andor Technology, Belfast, United Kingdom).

The LIBS spectra are represented in terms of intensity versus wavelength. Calibration of the spectrometer for accuracy is important in LIBS analysis. In the present study, wavelength and intensity calibrations of the Mechelle spectrograph, equipped with intensified charge coupled device (ICCD), utilize National Institute of Science and Technology (NIST) certified standard lamps. Two different types of standard lamps were used: (i) Hg-Ar (HG-1, Ocean Optics, Dunedin, FL, USA) lamp for wavelength calibration and (ii) deuterium-tungsten-halogen lamp (DH-2000-BAL, Ocean Optics) for intensity calibration.

The spectrometer was equipped with an ICCD (iStar 734, Andor Technology, Belfast, UK) and the obtained spectra were recorded with the Andor Solis software. The identification of lines was performed by matching the spectral lines and relative intensities using the NIST atomic spectroscopy database [23]. The gate delay and the gate width of the spectrometer were kept at 1 μs and 2 μs, respectively, for optimization of signal-to-noise and signal-to-background ratios.

In Raman spectroscopy for the molecular analysis of the sample, Raman microscopy measurements were performed with a Raman spectrometer (RIAR-532 Research India, having resolution 6 $cm^{-1}$). The Raman spectra were analyzed with reported data from Socrates [18] and Movasaghi et al. [25].

Fourier transform infrared (FT-IR) spectra were recorded using a spectrometer Spectrum-65 (Perkin Elmer) with a frequency range of 500–4000 $cm^{-1}$. The FT-IR spectroscopic analysis of each drug sample was carried out to evaluate the structure information with the help of literature [28,29].

In the present study, different brands of sample (named as S1, S2, S3, S4, S5, and S6) were used, the details of which are tabulated in Table 1. The LIBS spectra was recorded for 50 accumulations to produce one single, average LIBS spectrum. For each sample, 10 spectra were recorded. On the basis of spectral intensity variation of the ingredients of the drug, PCA was applied to the LIBS spectral dataset of drug samples [10].

## 4. Discussion and Conclusions

The simultaneous monitoring of atomic as well as molecular variations in drugs at the time of manufacturing can be quite challenging. This work introduces a new approach for drug compositional analysis, namely a combination of analytical methods including LIBS, and Raman and FT-IR spectroscopies. The study conditionally enables one to engage in simultaneous study of atomic and molecular compositions of pharmaceuticals using LIBS. The spectral signature of organic and inorganic elements can be detected with LIBS. The results reveal that the matrix of the pharmaceutical sample is composed of organic and inorganic materials. In addition to this, vibrational bands of $C_2$ Swan and CN violet band systems are also observed. The CN and $C_2$ band spectral intensities in LIBS spectra allow one to infer the presence of organic compounds/molecules in each sample [30]. In addition, complementary analysis of drugs is performed with Raman spectroscopy to determine the molecular composition of the drug. The observed spectral signatures in the Raman spectra are correlated with CN and $C_2$ bands intensity that are observed in the LIBS spectra.

Hardness testing may be important for manufacturing pharmaceuticals. This work computed the relative hardness of the investigated drugs. The chemometrics analysis using PCA of the recorded LIBS data suggests this application as a rapid and selective technique for the classification of selected pharmaceutical materials. Therefore, LIBS can be an effective tool for the study of compositional

variations in the form of trace elements, active and inactive ingredients, and molecules contained in the drug.

The presented results successfully demonstrate that LIBS is useful for elemental and molecular analysis. On the basis of our present work, one can say that LIBS along with chemometrics has good potential for the development and implementation of instruments for in-situ and online analysis of multiple elements, and molecular analysis of drugs [30].

**Author Contributions:** P.K.T. analyzed the samples following collection of experimental data, and he prepared the initial draft of this article. N.K.R. contributed in the analysis of Raman data and assisted in critical review of the manuscript. R.K. also focused on visualization of the outcomes, and he conducted the experiments jointly with P.K.T. C.G.P. suggested various approaches for the experimental studies and communicated substantially in the creation of the manuscript in its current form. A.K.R. motivated the work together with different approaches of drug analysis, and he provided expert advice in the interpretation and data analysis, moreover, A.K.R. provided significant support for this work.

**Funding:** This research received no external funding.

**Acknowledgments:** One of the authors, Pravin Kumar Tiwari, is thankful to Kuldeep Kumar Patel from Research India for providing the Raman experimental facility, and to Praveen Kumar Shahi, Assistant Professor Physics Department SPM degree college, University of Allahabad, Allahabad and S. B. Rai, Physics Department, Banaras Hindu University, Varanasi for FT-IR experimental support. Pravin Kumar Tiwari also thanks for support by the Board of Research in Nuclear Sciences, India, by providing Senior Research Fellowship support for the work project, Sanction No. 39/14/30/2016-BRNS/34434, 24.01.17.

**Conflicts of Interest:** There are no conflict of interest.

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
