# Peer review of "Atomic and Molecular Laser-Induced Breakdown Spectroscopy of Selected Pharmaceuticals"

_atoms, doi:10.3390/atoms7030071_

Round 1

Reviewer 1 Report

See attached PDF file for detailed comments

Author Response

Comments:

Fig. 1(a) needs to be “redrawn.”

Line 21: Insert “to” so that now reads “...used to record...”

Line 41: Omit “The” so that now reads “... [5-9]. Recent studies ...”

Lines 46 and 274: Change “in-situ” to “in situ” (italicized and no hyphen)

Line 58: Change “Another spectroscopic technique” to “Other spectroscopic techniques”

Line 59: Change “has” to “have”

Line 71: Change “pharmaceutical drug samples have” to “pharmaceutical pain-reliever drug

samples (Table 1) have”

Line 73: Change “component analysis” to “Component Analysis”

Response:

We have incorporated changes according to the suggestions.

Comments:

Lines 71 and 78: Move Table 1 closer to first mention in the paper. Currently Table 1 does not

occur until 4th page of paper; it should be on 2nd page of paper. Readers need to know which

drugs are being tested and how they differ from one another in order to understand how and

why the results are different for different samples; this needs to be presented as soon as

discussion of the results begins.

Response:

We agree with the learned reviewer’s comment and incorporated changes according to the suggestions.

Comments:

Lines 80-81: Change “The spectral lines” to “Spectral lines”. Insert comma after “elements”

Line 83: Insert comma and “also” so that now reads “... 288.1 nm), have also been ...”

Lines 85, 89, and 90: Insert “these” so now reads “... of these drugs.”

Line 87: Insert “the” so now reads “... of the drug samples”

Lines 88-89: Change “laser induced” to “laser-induced”

Line 92: Insert “(primarily)” so that now reads “... likely (primarily) due to ...”

Line 94: Change “sequence” to “sequences”

Line 97: Change “... in sample S6 [26].” to “... in sample S6, which contains Ibuprofen, but no

paracetamol [26].”

Response:

We have incorporated changes according to the suggestions.

Comments:

Line 100: Change “correspond” to “corresponding”.

Note the bands listed on this line are not shown [except in Fig. 1(a), where because of the vertical compactness of the spectra, it is difficult to compare one sample with another.

Line 101: Change “... bands [22].” to “... bands are observed in the LIBS spectra [22].”

Response: We have incorporated changes according to the suggestions and added Figure 1 (c) which represents the presence of mentioned bands in the LIBS spectra.

Comments:

Figure 1: Move text so that both Fig. 1(a) and Fig. 1 (b) [i.e., all of Fig. 1] is on the same page.

Response:

We have incorporated changes according to the suggestions.

Comments: If you compare “S6 spectrum” in Fig. 1(a) with “S6 spectrum” in Fig. 2(a), you will note that

the CN violet bands appear in the “S6 spectrum” of Fig. 1(a), but NOT in the “S6 spectrum”

of Fig. 2(a). Line 97 states that the CN violet bands are absent in S6 LIBS spectra; CN bands

should either be present in all S6 LIBS spectra OR should be absent in all S6 LIBS spectra.

Therefore, either “S6 spectrum” of Fig. 1(a) is NOT a LIBS spectrum of S6 OR “S6 spectrum”

of Fig. 2(a) is NOT a LIBS spectrum of S6.

If you consider the CN violet bands in Fig. 2(a), you note that although they differ in intensity,

the band positions are NOT significantly different from one sample to another. In contrast,

the CN violet band positions in Fig. 1(a) differ significantly from one sample to another (the

CN bands are at shorter wavelengths in the spectra of “S6”, “S5”, and “S1” than they are in the

spectra of “S4”, “S3”, and “S2.”) The authors need to go back to the original data (or

alternatively record new LIBS spectra); verify that the spectra are of the samples they claim;

and then completely “redraw” Fig. 1 (a) for all samples.

Response:

We agree with the learned reviewer’s comment and incorporated changes according to the suggestions.

Comments:

 Line 142: Change “... Ca II and of ...” to “... Ca II to that of ...” Ratio could either be ionic/neutral

OR neutral/ionic; need to be clear which ratio is meant.

Line 144: Change “... 279.5 nm, and of ...” to “...279.5 nm to that of ...”

Line 147: Change “... with sample [19-20].” to “... with sample can affect the intensities [19,

20].”

Line 149: Remove comma so now reads “... ionic lines to neutral lines ...”

Lines 150, 152, 174, 175, Table 2 caption, line 209: Change “&” to “and”

Line 152: Change “is” to “are”

Lines 164 and 168: Change “dataset” to “data set”

Lines 173-174: Omit “in” so now reads “... of different dimensions ...”

Line 185: Change “in-spite” to “inspite”

Line 186: Change “... in three dimensional.” to “... in three dimensions. PC-2 axis is

perpendicular to the plane of this page.”

Line 190: Change “... have firstly taken sample S1 and S3 ...” to “... have taken samples S1 and

S3 ...”

Line 193: Change “Analysis with Raman characteristic group frequencies [18] the peaks ...” to

“Analysis using Raman characteristic group frequencies [18] indicates that the peaks ...”

Line 194: Change “... mode, 1647 ...” to “... mode; and 1647 ...”

Line 197: Change “... to associate with ...” to “... to be associated with ...”

Lines 201 and 215: Change “Spectra” to “spectrum” in both instances in each line.

Line 203: Change “Sample” to “samples”

Line 206: Change “...spectrum of sample ...” to “spectra of samples ...”

Line 210: Change comma to semi-colon

Line 218: Change “spectroscopy” to “spectroscopies”

Line 221: Change “In LIBS set-up as described elsewhere [10], the high ...” to “In our LIBS set-

up as described elsewhere [10], high ...”

Line 222: Change “... laser device have been used to generate plasma ...” to “... laser have been

used to generate a plasma ...”

Line 234: Change “optics” to “Optics”

Line 236: Change “technology” to “Technology”

Line 240: Change “... signal to noise and signal to background ratios.” to “... signal-to-noise and

signal-to-background ratios.”

Line 243: Change “was” to “were”

Lines 243-244: Change “... Socrates et al. (2011) [18] and Zanyar Movasaghi et al. (2007) [24].”

to “... Socrates [18] and Movasaghi et al. [25].” In line 243, you list the year of publication of

Socrates as 2011, but in ref. 18, you list the year as 2001. Movasghi et al. is ref. 25, not ref. 24.

Lines 245-246: Change “... using the spectrometer Spectrum-65, (Perkin Elmer) with ...” to “...

using a Spectrum-65 spectrometer (Perkin Elmer) with ...”

Line 248: Insert parentheses to now reads “... sample (named as S1, S2, S3, S4, S5, and S6)

have ...”

Line 256: Change “Raman-“ to “Raman”

Line 257: Change “spectroscopy” to “spectroscopies”

Line 261: Change “intensity” to “intensities”

Line 262: Change “... spectra allows one to infer presence ...” to “... spectra allow one to infer

the presence ...”

Line 315: Change “et al.” to “M. Blanco, D. Moyan, N.W. Broad, N. O’Brien, D. Friedrich,

F.Pfeifer, and H.W. Siesler”

Line 371: Change “”Identification of molecular spectra London: Chapman and Hall”, (1976)” to

“”Identification of Molecular Spectra,” London: Chapman and Hall (1976)”

Line 381: Change “Christian G. Parigger et al.” to “C.G. Parigger, A.C. Woods, D.M. Surmick,

G. Gautan, M.J. Witte, and J.O. Hornkohl”

Response:

We have incorporated changes according to the suggestions.

Reviewer 2 Report

The paper is well written and covers an important topic of LIBS application in pharmacology. The paper demonstrates that atomic and molecular emission signatures from LIBS spectra can be used to determine material composition and properties in various samples. While LIBS techniques were very successful in using the line positions in material identification, line ratio analysis may pose certain challenges. Line ration, especially from different ionization stages of the same element, depend strongly on the temperature distribution in the ablated plume. In laser-produced plasmas, temperature gradients can be very significant. Details of the distribution can be affected by a number of the experimental conditions. Some discussion about potential problems with the line ratio analysis would be useful.  Also, it would be interesting to read why the authors selected PCA for the data examination.

The paper would definitely benefit from additional proof-reading.  For example, in the abstract one can find “are used record molecular spectra” and “The applicationof…”. Plot orientation in Figure 5 makes it difficult to interpret.

Author Response

Comments:

The paper would definitely benefit from additional proof-reading.  For example, in the abstract one can find “are used record molecular spectra” and “The applicationof…”. Plot orientation in Figure 5 makes it difficult to interpret.

Response:

We have incorporated several changes in the manuscript (green highlighted) to make it effective. Principle Component Scatter (PCA) plot of the drugs is represented in three dimensions (PC-1, PC-2 and PC-3). PC-2 axis is perpendicular to the plane of these page. This analysis will be significant enough in drug classification/discrimination when there are lot of samples of having similar compositions larger in number.
